# A Review of Recent Advances on Deep Learning Methods for Audio-Visual Speech Recognition

**Denis Ivanko** [†] , **Dmitry Ryumin** *,[†] and **Alexey Karpov**

St. Petersburg Federal Research Center of the Russian Academy of Sciences (SPC RAS),
199178 St. Petersburg, Russia; ivanko.d@iias.spb.su (D.I.); karpov@iias.spb.su (A.K.)
* Correspondence: ryumin.d@iias.spb.su
† These authors contributed equally to this work.

**Abstract:** This article provides a detailed review of recent advances in audio-visual speech recognition (AVSR) methods that have been developed over the last decade (2013–2023). Despite the recent success of audio speech recognition systems, the problem of audio-visual (AV) speech decoding remains challenging. In comparison to the previous surveys, we mainly focus on the important progress brought with the introduction of deep learning (DL) to the field and skip the description of long-known traditional "hand-crafted" methods. In addition, we also discuss the recent application of DL toward AV speech fusion and recognition. We first discuss the main AV datasets used in the literature for AVSR experiments since we consider it a data-driven machine learning (ML) task. We then consider the methodology used for visual speech recognition (VSR). Subsequently, we also consider recent AV methodology advances. We then separately discuss the evolution of the core AVSR methods, pre-processing and augmentation techniques, and modality fusion strategies. We conclude the article with a discussion on the current state of AVSR and provide our vision for future research.

**Keywords:** audio-visual speech recognition; lip-reading; audio-visual fusion; computer vision; deep learning; review

**MSC:** 68T10

## 1. Introduction

During the last decade, automatic speech recognition systems (ASRs) have already reached a certain performance level and are used in a number of practical applications [1–4]. However, sometimes, the quality and reliability of modern ASRs in real-world applications remain insufficient. Natural human speech is perceived by the listener both through acoustic and through visual channels (modalities of communication). Oral speech is produced multimodally and transmitted simultaneously through several channels [4], mainly as acoustic and visual information. The two sources of information complement each other and help to correctly understand speech in many difficult situations, e.g., in acoustically or visually noisy conditions [4]. There are several reasons why the visual modality improves human speech perception. For example, it facilitates speaker localization, it contains additional complimentary information that supplements the audio part, and it provides articulation information [5].

In order to improve the accuracy and reliability of ASR systems in the late 1990s, in addition to analyzing the information from only the acoustic modality, researchers began to use video information regarding speech. In many research works, the developed audio-visual speech recognition (AVSR) systems have demonstrated better recognition results [6–9] than their unimodal counterparts. Fusing streams of audio and visual modalities significantly increases the efficiency of ASR (i.e., the principle of synergy is respected). This fact is also known as the McGurk effect [10].

The above facts have generated considerable interest in VSR, officially known as automatic lip-reading. Not surprisingly, the inclusion of an additional visual modality has resulted in an increase in the performance of audio-only ASR over a wide range of conditions. Speech is the most natural, convenient and understandable way for people to communicate. However, modern ASR systems often exhibit reduced performance in real-world, noisy environments due to the discrepancy between controlled laboratory training procedures and real-world environmental conditions. In recent years, many technologies have been created to achieve noise robustness [11]. However, most of these technologies cannot provide this in real conditions with various types of noise [11]. However, visual information is not distorted by acoustic noise, and automatic lip-reading plays an important role in acoustically difficult environments [11]. The additional visual modality greatly improves performance, particularly in noisy environments where traditional acoustic ASRs struggle.

AVSR systems have gained significant traction in the field of ML due to their central role in a wide range of emerging applications. This fundamental and challenging topic has received increasing attention in recent years. AVSR systems can be applied to a much wider range of application scenarios, such as command recognition in vehicles [12,13], mobile phone text dictation [12,14], lip-reading for the hearing impaired [15–18], speech recognition of individual speakers from multiple people speaking at the same time [19], etc.

Despite the significant advances in DL that have been made in recent years in a wide range of fields, including computer vision (CV) (semantic segmentation [20–24], scene understanding [25–29], pose estimation [30–36], action [36–38] or gesture [39–43] classification, face [44–47] or emotion [48–51] recognition, etc.), natural language processing (text analysis [52–54], language translation [55–57], sentiment analysis [58–60], question answering [61], etc.), speech recognition [62–65], and generative design (automated content generation [66], creative image synthesis [67–69], artistic style transfer [70–72], etc.), the development of AVSR methodology is still at an early stage and does not yet meet the performance standards required for practical implementation in real-world applications. This is certainly not due to a lack of effort on the part of researchers, as there have been many excellent works on AVSR [73]. Therefore, a comprehensive analysis of recent advances, identification of key barriers and unresolved issues, and exploration of potential avenues for future research are essential. However, existing research on AVSR lacks coherence, and systematic reviews addressing these aspects are currently lacking. This review aims to fill this gap and provide a comprehensive overview. We provide a comprehensive review that covers aspects of modern state-of-the-art (SOTA) AVSRs.

This article focuses on the most recent advances in audio-visual (AV) speech decoding. The recent surge in DL has led to significant advances in the field. Our main intention is to supplement (but not replace) previous studies [5,74–76]. We propose to look at the problem of AVSR from the DL point of view. We do not consider so-called "traditional approaches" associated with hand-crafted features, Markov models used for recognition [75], etc. Additionally, we focus more on modern end-to-end (E2E) AVSR systems and related methodologies.

In this article, we present the main techniques for AVSR that have been developed over the last decade (2013–2023). First, in Section 2, we discuss the main AV datasets used in the literature for AVSR experiments since we consider it a data-driven machine learning (ML) task. Then, in Section 3, we consider the methodology used for visual speech recognition (VSR). Subsequently, in Section 3, we also consider recent AV methodology advances. In Section 4, we discuss the evolution of the core AVSR methods, pre-processing and augmentation techniques, and modality fusion strategies. Finally, in Section 5, we conclude the article with a discussion on the current state of AVSR and provide our vision for future research.

## 2. Recent Audio-Visual Speech Datasets

The design and development of modern AVSR systems is a data-driven process and is inevitably affected by the amount and quality of available data. In this section, we review the recent AV speech datasets between 2013 and 2023, highlighting the progression from rather small traditional datasets toward large-scale datasets to train E2E deep learning (DL) architectures.

While there exist several other literature reviews on visual and AV speech datasets, some of them are already outdated, e.g., [74], which only covers datasets up to 2013, or [77], which covers datasets up to 2018. The significant growth of research in visual and AVSR has considerably expanded the literature of the field, pushing forward the SOTA systems toward DL architectures and justifying the need for an up-to-date review [78] that covers only recent advances without going deep into outdated legacy approaches.

On the other hand, some other reviews, such as [73,79], cover more recent datasets. However, these works provide only a combined list of datasets in one table and focus mainly on lip-reading datasets rather than on AV datasets.

In our survey, we provide a comprehensive list of the existing datasets and categorize them into two different groups. The first group includes large-scale publicly available datasets used for AVSR benchmarking, with each dataset used in at least 10 other research works.

The second group includes datasets that are collected in laboratory conditions and are specifically designed for certain tasks, e.g., analysis of the influence of frame rate, analysis of certain languages other than English, analysis of the Lombard effect, etc. The second group is united by the fact that often, the AVSR results are given only by the authors of the dataset. We also include some large-scale datasets in this group, for which the main results are presented only by the authors.

In addition to this categorization, we describe the popularity of the datasets, what tasks they can be used for, as well as their most important characteristics, such as their number of speakers [77], vocabulary size [77], recording settings [77], duration [77], language, year of creation, etc.

### 2.1. Benchmarking Datasets

Datasets have played a very important role throughout AV speech research development, especially in the big data era. In the first group, we include the AV speech datasets that are most popular among researchers for benchmarking AVSR or lip-reading systems. Benchmark datasets provide a common platform to measure and compare the performance of SOTA algorithms. DL technologies have made significant progress in AVSR in recent years, but it is worth noting that the appearance of large-scale datasets with annotated data play a crucial role in their success. Here we describe 7 [80–86] publicly available large-scale datasets suitable for training DL architectures with at least 10 research groups reporting results on each dataset. Their main characteristics are shown in Table 1.

**Table 1.** Main characteristics of the benchmarking datasets. UNK is unknown.

| Name | Year | Speakers | Vocabulary Size | Word Instances | Total No. of Utterances | Duration, Hours | Resolution | FPS | Audio, kHz | View |
|------|------|----------|-----------------|----------------|-------------------------|-----------------|------------|-----|------------|------|
| LRW [80] | 2016 | 1000+ | 500 | ~539,000 | ~539,000 | 173 | 256 × 256<br>112 × 112 | 25 | 16 | Front |
| LRS [82] | 2017 | 1000+ | 17,428 | ~782,000 | 118,116 | 75.5 | 256 × 256 | 25 | 16 | Front |
| MV-LRS [85] | 2017 | 1000+ | 14,960 | ~470,000 | 74,564 | 155 | 224 × 224 | 25 | 16 | Profile |
| LRS2-BBC [83] | 2018 | 1000+ | 60,693 | ~2,343,663 | 144,482 | 224.5 | 224 × 224 | 25 | 16 | Front |
| LRS3-TED [84] | 2018 | 9279 | 71,136 | ~4,561,000 | 165,452 | 475 | 224 × 224 | 25 | 16 | Front |
| LSVSR [86] | 2018 | 1000+ | 127,055 | ~350,000 | 2,934,899 | 3886 | 128 × 128 | 23+ | UNK | Front |
| LRW-1000 [81] | 2019 | 2000+ | 1000 | – | – | – | Different | 25 | 16 | Front |

The acquisition of large AV speech datasets in a real environment is challenging due to many factors, such as the subjects, illumination, noise, head-pose, vocabulary, resolution, etc. The lack of suitable datasets has been one of the major obstacles to progress in the AVSR

field for a long time. However, the total duration of AV information was far from sufficient to train representative DL-based models that are able to generalize beyond the laboratory conditions and very limited vocabularies until the introduction of the first large-scale lip-reading in-the-wild dataset (LRW [80]) in 2016 nearly 7 years ago.

The revolutionary multi-stage pipeline proposed in [80] allows for the automatic collection and processing of very large-scale datasets based on open-source television programs. It generally includes 5 processing stages, namely, (1) selecting program types; (2) subtitle processing and alignment; (3) shot boundary detection, face detection, and tracking; (4) facial landmark detection and speaker identification; and (5) compiling the training and test data. Using this pipeline, the authors collected a famous LRW dataset. It is worth noting that all other benchmarking datasets presented in Table 1 were collected and annotated using a similar processing pipeline. Thus, we can conclude that 2016 was a turning point in changing the approach to creating AV speech datasets from laboratory conditions to real conditions.

The authors in [80] were the first to collect training data for several hundreds of words with thousands of instances per word and over a thousand different speakers. The dataset was collected from British television programs, with each video corresponding to a word category. All the videos in this dataset are of fixed duration, size, and length [81]. It provides much convenience to the research community and is especially useful due to the fact that neural network (NN) inputs usually demand batches of fixed size. This remains a challenging dataset [81] at present and has been widely used by most existing lip-reading and AVSR methods.

However, one default characteristic of the LRW dataset is that all the words are forced to have an equal duration, and each class contains the same number of samples (balanced classes). This creates a mismatch between the collected data and real-life applications, as word frequencies and speech rates in real life scenarios differ from those observed in the collected data.

Based on this conclusion, the authors in [81] proposed a naturally distributed modification to the LRW dataset and gave it the name LRW-1000. The dataset contains large variations in speech conditions, lighting conditions, video resolution, speaker pose, speech rate, age, gender, etc. It is quite unbalanced due to some classes containing more samples than others due to certain words naturally occurring far more often than others. In comparison to traditional LRW, the video samples are not limited to a specific length. Overall, the dataset [81] is very impressive, showing extensive variation in speaker characteristics, such as pose, age, and gender, as well as scale, video resolution and background conditions.

Unlike the LRW or LRW-1000 datasets, which focus on word recognition tasks, the lip-reading sentences in-the-wild dataset (LRS) was the first to tackle an unconstrained open-world sentence recognition task in [82]. It consists of video sentences from British television. Along with its introduction, it was demonstrated that a current SOTA lip-reading system could beat a professional lip reader on videos from BBC television, and if audio was available, it was found that visual information still helped to improve speech recognition performance. LRS contains 17,428 different words combined in 118,116 utterances [77] and at the time of release was the largest AV dataset.

Following the success of the initial LRS dataset, in 2018, two other similar datasets were released: LRS2-BBC [83] and LRS3-TED [84]. The authors of LRS2-BBC used a wide range of BBC programs to collect thousands of hours of spoken sentences and phrases, accompanied by their corresponding facetracks. The vocabulary size of LRS2-BBC exceeds LRS over 3 times (75 k to 25 k), the total amount of utterances exceeds LRS by 26 k (144 k to 118 k), and the total duration exceeds LRS by almost 3 times (224.5 h to 75.5 h).

Because the LRW, LRS, and LRS2-BBC datasets are subject to some restrictions and can only be used for scientific research labs and not by industrial companies, in 2018, researchers released a new LRS3-TED dataset to provide a common benchmark dataset for all type of researchers [84]. It consists of over 400 h of AV data extracted from 5594 English TED talks downloaded from YouTube [87], which makes it more than twice as big as the previous

LRS2-BBC dataset (475 h to 224.5 h). It also exceeds LRS2-BBC in the total number of utterances (165 k to 144 k) and in the amount of vocabulary words (71 k to 60 k).

The only profile-view dataset on our list is MV-LRS [85], which was introduced in 2017 in an attempt to answer the question of whether it is possible to lip-read from a profile with the same accuracy as from frontal or near frontal faces. To date, MV-LRS is the largest AV speech dataset that contains profile faces, with over 777 h of speech data. At the same time, the vocabulary (14 k to 17 k) and number of utterances (74 k to 118 k) are smaller than in the LRS dataset. In contrast to the LRS dataset, which primarily consists of videos from frontal news broadcasts, MV-LRS includes a broader range of programs, including dramas and factual programs with conversational interactions and sideways portraits of individuals.

The large-scale VSR dataset (LSVSR) collected by the Google team was presented in Ref. [86] and consists of video clips of faces speaking derived from 3886 h of YouTube videos. To date, it is the largest reported dataset; however, it is not publicly available. Thus, only the baseline AV recognition results presented in the article are available.

All the datasets presented in the first group were chosen as benchmarks by the AVSR research community, and now, we can see the reason for this. All the presented datasets share several common features, such as over 1000 different speakers each, which inevitably leads to large vocabulary sizes, a large number of utterances, and longer durations. All the data were collected via the internet, either from TV programs, YouTube, etc., and contain uncontrolled speech without specific environments, scripts, etc. Another interesting feature is that the researchers prefer rather low-resolution videos with a standard 25 fps frame rate and 16 kHz of audio discretization. The obvious explanation for this is that such videos take up much less space than HD or FullHD, allowing to fit more videos in the same amount of processing and storage memory. Thus, we can conclude that modern big-data-era AV datasets have the largest possible vocabulary and speaker variability, despite, to some extent, sacrificing the high quality of AV data. It is also worth mentioning that all first-group datasets use the English language as their source data.

*2.2. Research Datasets*

The second group is composed of the research datasets, which are usually collected in controlled laboratory conditions to address a specific AVSR problem, issue, component of vocabulary, etc. In addition, most of these datasets were proposed by separate research groups and are not publicly available. Thus, usually these datasets have only one baseline result published by the authors. Table 2 lists some of the main AV datasets that have been presented for AVSR or lip-reading over the last 10 years in chronological order.

The AVAS [88] dataset is the first Arabic AV speech dataset. It was recorded in an office environment under different acoustic conditions, four different illumination conditions, and with five head pose variations for each speaker. It is useful for three research tasks, Arabic AVSR, multi-view lip-reading, and noise (acoustic and visual) robustness. The speech materials are composed of 36 daily words and 13 casual phrases recorded under a controlled environment by 50 different speakers.

The AusTalk [89] dataset is the first Australian English dataset. It was designed with the two main goals of providing a standardized infrastructure for AV recordings in Australia and of producing a large AV dataset. At the time of release, it consisted of 3 h of AV recordings for 1000 speakers. The authors claim that the dataset has been expanded significantly by 1000 times and currently consists of 3000 h of recordings of Australian English. However, it is not publicly available. The AusTalk dataset is definitely useful for the task of Australian English AVSR and if, ever released, suitable for training modern DL architectures.

The TCD-TIMIT [90] dataset consists of high-quality AV data of 62 speakers reading a total of 6913 sentences. In contrast with benchmarking datasets from the first group, which usually sacrifice video resolution for processing efficiency, the TCD-TIMIT dataset has FullHD video resolution of 1920 × 1080 pixels. This dataset is free and publicly available and is one of the most popular among researchers. Another interesting feature of this

dataset is that three of the speakers are professionally trained lip speakers. Thus, there is a human benchmark available to challenge the automatic AVSR systems.

**Table 2.** Task-specific AV datasets: **S**—speakers, **D**—duration, Ar—Arabic, Au—Australian, En—English, Ru—Russian, Es—Spanish, Zh—Chinese, Tr—Turkish.

| Name | Year | S | Vocabulary Instances | Total No. of Utterances | D, Hours | Resolution | FPS | Audio, kHz | Language | Gender Male/Female | View |
|---|---|---|---|---|---|---|---|---|---|---|---|
| AVAS [88] | 2013 | 50 | 47 | 13,850 | – | 640 × 480 | 30 | – | Ar | – | Angles |
| AusTalk [89] | 2014 | 1000 | – | – | 3000 | 640 × 480 | – | – | Au En | – | Front |
| TCD-TIMIT [90] | 2015 | 62 | – | 6913 | 6 | 1920 × 1080 | 30 | 16 | En | 32/30 | Front |
| OuluVS2 [91] | 2015 | 53 | – | 20,000+ | ~2.25 | 1920 × 1080 | 30 | 44.1 | En | 40/13 | Angles |
| | | | | | | 640 × 480 | 100 | | | | |
| IBM AV-ASR [92] | 2015 | 262 | 10,400+ (42 ph) | – | 40 | 704 × 480 | 30 | 16 | En | – | Front |
| | | | | | | 64 × 64 | | | | | |
| HAVRUS [93] | 2016 | 20 | 200 | 4000 | - | 640 × 480 | 200 | 44.1 | Ru | 10/10 | Front |
| MODALITY [94] | 2017 | 35 | 182 | 5880 | 31 | 1920 × 1080 | 100 | array | En | 26/9 | Front |
| VLRF [95] | 2017 | 24 | 1374 | 10,200 | 3 | 1280 × 720 | 50 | – | Es | – | Front |
| Mobio [96] | 2017 | 150 | – | – | – | 640 × 480 | 16 | – | En | – | Front |
| AV Lombard GRID [97] | 2018 | 54 | 100 | 16,200 | – | 720 × 480 | 24 | 16 | En | 24/30 | Angles |
| AV Digits [98] | 2018 | 53 | 10 + 10 | 6695 | – | 1280 × 780 | 30 | 44.1 | En | 41/12 | Angles |
| CMLR [99] | 2019 | 11 | 3517 | 102,072 | - | 64 × 128 | – | – | Zh | – | Front |
| AVSD [100] | 2019 | 22 | 10 | 1100 | – | 1920 × 1080 | 30 | – | Ar | – | Front |
| HDTF [101] | 2020 | 300+ | – | 10,000+ | 16 | 256 × 256 | – | – | En | – | Front |
| VR Digits [102] | 2020 | 6 | 10 | 6000 | – | 1920 × 1080 | 25 | – | En | – | Front |
| NSTDB [103] | 2020 | – | 349 | – | – | 64 × 64 | 25 | – | Zh | – | Profile |
| LRWR [104] | 2021 | 135 | 235 | – | – | 112 × 112 | 25 | – | Ru | – | Front |
| RUSAVIC [105] | 2022 | 20 | 134 | 26,800 | – | 1920 × 1080 | 60 | 48 | Ru | 10/10 | Front |
| VLRDT [106] | 2023 | – | 10 | 2335 | – | – | – | – | Tr | – | Front |
| CN-CVS [107] | 2023 | 2500 | – | 200,000 | 300 | – | – | – | Zh | – | Front |
| MuAViC [108] | 2023 | 8000+ | – | – | 1200 | – | – | – | 9 languages | – | Front |

The OuluVS2 [91] multi-view dataset, similar to TCD-TIMIT, has a high resolution of 1920 × 1080 pixels on one camera and a smaller resolution of 640 × 480 on a high-speed 100 fps camera. It has 52 speakers and nearly 1600 utterances and can be used for research on three tasks: the influence of different camera resolutions on speech recognition accuracy, the influence of different frame rates (25–100 fps) on AVSR accuracy, and multi-view lip-reading tasks.

The IBM AV-ASR [92] dataset consists of 40 h of AV recordings from 262 speakers and more than 10 k words, but unfortunately, it is not publicly available. If ever released, the dataset would be suitable for training modern DL architectures. In its current state, it is of limited interest to AVSR researchers.

The HAVRUS [93] dataset consists of high-speed (200 fps) video camera recordings of speech in Russian language. In total, the utterances of 20 speakers (10 male and 10 female) were collected during the recordings. Each of the speakers uttered 200 phonetically rich phrases. The HAVRUS dataset has the highest video frame rate among all the existing AV speech datasets and is of great interest in researching the influence of different frame rates on AVSR accuracy.

The MODALITY [94] dataset contains 31 h of recordings and is created specifically to assist the development of AVSR systems. The dataset includes high-resolution, high-frame-rate AV data. The dataset includes recordings of 35 speakers. Every utterance was manually labeled, resulting in label files being added to the dataset repository [94].

The VLRF [95] dataset was collected for the purpose of speech recognition in the Spanish language, mainly for visual-only speech recognition. In total, there are 10,200 words and phrases, with an average duration of 7 s per sentence. The overall dataset duration is 180 min. The sentences exhibit a well-balanced phonological distribution representative of the Spanish language.

The Mobio [96] dataset consists of AV data collected from 152 speakers. The dataset has 100 males and 52 females and was collected in six different cities from five different countries. It includes both native and non-native speakers of the English language. The dataset was recorded using two devices: a smartphone and a portable computer. In total,

12 sessions were recorded for each participant. The Mobio dataset was primarily recorded for mobile face and speaker recognition but can also be used in AVSR tasks.

The AV Lombard GRID [97] is a multi-view AV Lombard speech dataset that was designed to research behavioral studies in speech perception. The dataset consists of recordings of 54 speakers. It has 100 phrases per speaker (50 Lombard and 50 normal utterances). It offers two different views of the participants (frontal and side views) to facilitate analysis of speech from different angles. Currently, it is the only AV speech dataset designed for the investigation of the Lombard effect in AV speech.

The AV Digits [98] dataset is a distinctive speech dataset that encompasses normal, whispered, and silent speech. It consists of two parts: digits and short phrases [98]. The dataset includes recordings from 53 participants representing 16 nationalities, with 41 males and 12 females. The speaker's face was captured from three perspectives using three cameras with frontal, $45°$, and profile views. The audio was simultaneously recorded by the built-in microphones of the cameras at a sampling rate of 44.1 kHz.

The CMLR [99] is an AV speech dataset consisting of 102,072 spoken phrases in Chinese produced by 11 participants. It was recorded between June 2009 and June 2018. Each phrase in the dataset [99] consists of approximately 29 Chinese characters and does not include any English letters, Arabic numerals, or uncommon punctuation marks. The sentences also provide the alignment boundaries of each word in seconds. It is one of the few benchmarks available for languages other than English.

The AVSD [100], or the Arabic Visual Speech Dataset, includes recordings of a modern standard Arabic (MSA) dataset for speech recognition systems and was acquired by one smartphone camera. The AVSD includes 1100 videos for 10 daily communication words uttered by 22 Arabic speakers [100]. The dataset was collected in diverse indoor environments, capturing realistic conditions and variations in room lighting.

The HDTF [101] dataset was designed for talking face generation rather than AVSR. It is a medium-size dataset with a higher-than-usual video resolution than can be found in previous in-the-wild datasets. It has more data/phrases than other in-the-lab datasets. The HDTF dataset consists of 362 different videos with a total duration of 15.8 h.

The VR Digits [102] dataset consists of recordings of 10 different digital phrases (numbers from zero to nine) and was collected from six speakers (3 males and 3 females). Each speaker uttered each sentence about 100 times. The original video resolution was $1920 \times 1080$ with a frame rate of 25 frames per second. After processing each video frame, the authors provided a region-of-interest (ROI) of $224 \times 224$ pixels as their standard.

The NSTDB [103] is a sentence-level Mandarin lip-reading dataset that contains 1705 characters of the Chinese language. Some of the characters appear only a few times only. This dataset is notably challenging because all of the classes are very unbalanced.

The LRWR [104] is a naturally distributed large-scale dataset for lip-reading in the Russian language that includes 235 words from 135 speakers, with over 117,500 samples in total. To the best of our knowledge, it is the first Russian lip-reading dataset collected in the wild from YouTube videos. One of the difficult challenges in the AVSR research field is the lack of properly sized datasets for a different language; to date, all state-of-the-art methods were initially proposed only for the English language. This dataset was created to eliminate this language gap.

The RUSAVIC [105] dataset was collected for a specific task of AV command recognition in acoustically noisy driving conditions. It includes recordings of 20 drivers with a minimum of 10 recording sessions [105]. Each speaker recited a script from three different dictionaries, including 62 of the most frequent requests from drivers to smartphones, 33 letters of the Russian alphabet, and 39 numbers (including tens and hundreds) [105]. Since the dataset was recorded in natural conditions, the average SNR varies from 30 to 5 dB [105]. The video resolution was FullHD resolution, with $1920 \times 1080$ pixels and a 60 fps recording rate [105].

The VLRDT [106] dataset contains a 2335 instances in the Turkish language taken from YouTube videos. The instances in the dataset consist of videos with different angles, shadows, resolutions, and brightnesses that are not manually generated [106].

The CN-CVS [107] dataset is a Mandarin Chinese AV dataset consisting of short snippets of human speech extracted from news broadcasts, TV shows, and web-based speech or conversation programs. It contains recordings from over 2500 speakers of different professions and ages. The dataset [107] is recorded in natural, uncontrolled environments where environmental factors such as lighting conditions may vary between programs or locations. The camera angle and distance also vary within the same video clips [107]. CN-CVS includes both audio and video components, with an average segment length of 6 s. The dataset [107] includes a wide range of content covering different subject areas.

The MuAViC [108] dataset is a pioneering multilingual AV dataset designed for robust speech recognition and speech-to-text translation. It contains 1200 h of AV speech data in nine different languages [108]. It is the first publicly available benchmark for AV speech-to-text translation and the largest open benchmark for multilingual AVSR. The dataset consists of transcribed AV speech from 8000+ speakers in TED and TEDx talks.

After carefully analyzing the existing AV speech datasets, we can state that task-specific datasets have a much greater variety than universally recognized benchmarking datasets. First of all, we are starting to see the diversity of languages. Of course, English speech datasets still predominate, but a large number of datasets for other languages are also emerging, including datasets for Arabic, Chinese, Russian, Spanish, and Turkish, as well as the MuAViC [108] dataset that contains the recordings of 9 different languages.

Another common feature of the abovementioned datasets is a rather small average number of speakers (20 to 100) in comparison to benchmarking datasets (1000+). Some rare exceptions here include AusTalk, IBM AV-ASR, and HDTF, which have many speakers but are unfortunately publicly unavailable. Thus, it is currently impossible to benchmark on these datasets. The CN-CVS and MuAViC datasets are also publicly available and have a sufficient number of speakers (1000+) to be on the benchmarking list. However, there are still no scientific works presented in the literature that use these datasets other than the actual authors of the datasets. That is probably due to the fact that both datasets were released just recently. These works definitely have a chance to be on the benchmarking list in upcoming years.

It is important to note that there is a lack of similar datasets specifically designed for this task. Each existing dataset has different characteristics, including variations in video resolution (ranging from 64 × 64 pixels to 1920 × 1080 pixels in the NSTDB, TCD-TIMIT and MODALITY datasets), recording speed (ranging from 15 fps to 200 fps in the MIRACLE-VS and HAVRUS datasets), and audio discretization (ranging from 16 kHz to 48 kHz in the TCD-TIMIT, IBM AV-ASR, and RUSAVIC datasets), as well as differences in vocabulary, total number of utterances, and duration.

## 3. Methodology

We evaluated the methodology of SOTA AVSR based on the selected list of benchmark datasets since the task of ML is highly data-dependent. This allows us to look at the problem and the choice of the necessary methods both in terms of their effectiveness and in terms of further practical applications because different methods will be most suitable for different types of data and tasks. This study provides an overview of the current state of the field in relation to the most popular methods and datasets. In this section, we evaluate the recent progress in AVSR methodology based on selected reference datasets. We determine which methods have become a recent turning point and have improved the accuracy of speech recognition both in speakers' lip-reading and AVSR tasks. The main goal is to determine which are promising and which are dead ends for further research. We start with the most popular dataset for assessing the accuracy of automated lip-reading (LRW), although it is worth noting that some AV results for this dataset are also available in the scientific literature.

### 3.1. LRW Methodology: Visual Speech Recognition

Table 3 shows the reported results of automatic lip-reading and AVSR accuracy based on the LRW dataset. The methods are listed in ascending order of reported recognition accuracy, not chronologically. LRW has been quite popular in recent years, and we were able to find 20 different methods that were proposed and used this dataset. It is worth noting that most of the proposed methods are suitable for assessing automatic lip-reading, and there is only one method for AV recognition. Therefore, LRW and the methods described below can be considered as references for automatic lip-reading rather than AVSR.

**Table 3.** AVSR methodology on LRW dataset. **V** is video, **A** is audio, and **AV** is audio-visual.

| No. | Method | Year | V | A | AV | Accuracy |
|-----|--------|------|---|---|----|----------|
| 1 | 3D Conv + ResNet-34 + Bi-LSTM [109] | 2017 | ✓ | – | – | 83.00 |
| 2 | Multi-grained + Bi-ConvLSTM [110] | 2019 | ✓ | – | – | 83.34 |
| 3 | 3D Conv + ResNet-34 + Bi-GRU [111] | 2018 | ✓ | – | – | 83.39 |
| 4 | PCPG [112] | 2020 | ✓ | – | – | 83.50 |
| 5 | DFTN [113] | 2020 | ✓ | – | – | 84.13 |
| 6 | SpotFast + Transformer + Product key memory [114] | 2020 | ✓ | – | – | 84.40 |
| 7 | 3D Conv + ResNet-18 + Bi-GRU [115] | 2020 | ✓ | – | – | 84.41 |
| 8 | 3D Conv + P3D-ResNet50 + TCN [116] | 2020 | ✓ | – | – | 84.80 |
| 9 | MoCo + Wav2Vec [117] | 2022 | ✓ | – | – | 85.00 |
| 10 | 3D Conv + ResNet-18 + Bi-GRU (Face Cutout) [118] | 2020 | ✓ | – | – | 85.02 |
| 11 | 3D Conv + ResNet-18 + MS-TCN [119] | 2020 | ✓ | – | – | 85.30 |
| 12 | 3D Conv + ResNet-18 + Bi-GRU + Visual-audio memory [120] | 2021 | ✓ | – | – | 85.40 |
| 13 | 3D-ResNet + Bi-GRU + MixUp + Label smoothing + Cosine LR [121] | 2020 | ✓ | – | – | 85.50 |
| 14 | 3D-ResNet + Bi-GRU + MixUp + Label smoothing + Cosine LR (Word Boundary) [121] | 2020 | ✓ | – | – | 88.40 |
| 15 | 3D Conv + ResNet-18 + MS-TCN + KD (ensemble) [122] | 2020 | ✓ | – | – | 88.50 |
| 16 | 3D Conv + ResNet-18 + MS-TCN + Multi-head visual-audio memory [123] | 2022 | ✓ | – | – | 88.50 |
| 17 | Vosk + MediaPipe + LS + MixUp + SA + 3DResNet-18 + BiLSTM + Cosine WR [13] | 2022 | ✓ | – | – | 88.70 |
| 18 | 3D Conv + EfficientNetV2 + Transformer + TCN [124] | 2022 | ✓ | – | – | 89.52 |
| 19 | 3D Conv + ResNet-18 + DC-TCN + KD [125] | 2022 | ✓ | – | – | 94.10 |
| 20 | 2DCNN + BiLSTM + ResNet + MLF [11] | 2023 | ✓ | ✓ | ✓ | 98.76 |

The first method (No. 1, Table 3) was proposed shortly after the release of a dataset in [109] in 2017 and demonstrated a recognition accuracy of 83.00%. In addition to general facial landmark detection and data augmentation, the proposed algorithm consists of three main parts: (1) a spatio-temporal front-end, (2) a residual network, and (3) a bidirectional long-short term memory (LSTM) back-end.

The main idea of the first set of 3DConv layers is to apply space–time convolution to a preprocessed frame stream. These spatio-temporal convolutional layers are able to capture the short-term dynamics of the mouth area [126]. Convolutional neural networks (CNNs) containing complex convolutions working in the image space have played an important role in improving performance in CV tasks. A basic 2D convolution layer from C channels to $C'$ channels (without a bias and with a unit stride) can be computed as [126]

$$[\text{conv}(\mathbf{x}, \mathbf{w})]_{c'ij} = \sum_{c=1}^{C} \sum_{i'=1}^{k_w} \sum_{j'=1}^{k_h} w_{c'ci'j'} x_{c,i+i',j+j'}, \tag{1}$$

for an input $\mathbf{x}$ and the weights $\mathbf{w} \in \mathbb{R}^{C' \times C \times k_w \times k_h}$, where we define $x_{cij} = 0$ for $i$ and where $j$ is out of bounds. Spatio-temporal convolutional neural networks (STCNNS) can process video data by convolutions in the time as well as in the spatial dimension. According to Ref. [127], we can have

$$[\text{stconv}(\mathbf{x}, \mathbf{w})]_{c'tij} = \sum_{c=1}^{C} \sum_{t'=1}^{k_t} \sum_{i'=1}^{k_w} \sum_{j'=1}^{k_h} w_{c'ct'i'j'} x_{c,t+t',i+i',j+j'}, \tag{2}$$

In this particular work, the authors considered a convolutional layer with 64 3-dimensional (3D) kernels of $5 \times 7 \times 7$ size (time/width/height), followed by batch normalization (BN) [128] and rectified linear units (ReLU).

The obtained 3D feature maps were then passed through a residual network, one per time-step. The authors used a 34-layer ImageNet version [129]. Its building blocks

were composed of two convolutional layers, followed by BN and ReLU, while the skip connections facilitated information propagation [130]. ResNet used max-pooling layers to progressively reduce the spatial dimensionality, resulting in a one-dimensional tensor per time step as the final output.

The authors used a bidirectional LSTM network as the back-end of the model. It consisted of two stacked LSTMs for each direction, and the final LSTM outputs were concatenated for combination. All these components became the core of modern SOTA AVSRs.

A slight improvement upon this baseline (No. 2, Table 3), with 0.34% better recognition accuracy, was achieved in [110]. The authors proposed an interesting concept of using 3 modules, namely a fine-grained module, a medium-grained module, and a coarse-grained module, followed by bidirectional ConvLSTM with forward input attention. The fine-grained module consisted of a 2D ResNet-34 network and was mainly used to observe slight differences in words with similar mouth movements. The medium-grained module consisted of a 52-layer 3D-DenseNet architecture that was used to capture short-term spatio-temporal patterns. This module was more accurate in processing motion information. The coarse-grained module was used to fuse features from the previous two modules with the help of an attention mechanism. This spatial attention mask and the final fused features $F$ were obtained by [110]

$$\mathbf{S} = 2\text{DCNN}(\mathbf{X}), \quad \mathbf{T} = 3\text{DCNN}(\mathbf{X}),$$
$$\text{mask} = \sigma(\mathbf{WT}), \tag{3}$$
$$\mathbf{F} = \mathbf{T} \odot \text{mask} + \mathbf{S} \odot (1 - \text{mask}),$$

where $X$ denotes the input feature maps, $S$; $T$ stand for the respective outputs of the two branches; $W$ is a learned parameter; $\sigma$ denotes the sigmoid function; and $\odot$ is an element-wise multiplication. The two-level bidirectional ConvLSTM module, complemented by attention to direct input, was used to model global hidden patterns throughout the sequence based on combined initial representations.

The next improvement to this 83.39% lip-reading accuracy (No. 3, Table 3) was achieved in [111]. The authors used a 34-layer ResNet for spatio-temporal processing; however, they changed the back-end model from Bi-LSTM to a bidirectional gated recurrent unit (GRU) with 1024 cells in each layer, which resulted in a slight recognition improvement. The GRU is an advanced variant of the RNN architecture. It overcomes the limitations of earlier RNNs by introducing additional cells and gates that allow information to be distributed over a greater number of time steps. This allows the GRU to effectively process and manage the flow of information within the NN. The standard formulation [111] is as follows:

$$[\mathbf{u}_t, \mathbf{r}_t]^T = \text{sigm}(\mathbf{W}_z \mathbf{z}_t + \mathbf{W}_h \mathbf{h}_{t-1} + \mathbf{b}_g),$$
$$\tilde{\mathbf{h}}_t = \tanh(\mathbf{U}_z \mathbf{z}_t + \mathbf{U}_h (\mathbf{r}_t \odot \mathbf{h}_{t-1}) + \mathbf{b}_h), \tag{4}$$
$$\mathbf{h}_t = (\mathbf{1} - \mathbf{u}_t) \odot \mathbf{h}_{t-1} + \mathbf{u}_t \odot \tilde{\mathbf{h}}_t,$$

where $\mathbf{z} := \{\mathbf{z}_1, \ldots, \mathbf{z}_T\}$ denote the input sequence, $\odot$ stands for element-wise multiplication, and $sigm(r) = 1/(1 + exp(-r))$. Bi-GRU ensures that $ht$ depends on $zt'$ for all $t'$. To parameterize a distribution over sequences, at time-step $t$, let $p(ut \mid z) = $ softmax(mlp($ht; Wmlp$)), where $mlp$ is a feed-forward network with weights $Wmlp$. Then, we can define the distribution over length-$T$ sequences as [111]

$$p(u_1, \ldots, u_T \mid \mathbf{z}) = \prod_{1 \leq t \leq T} p(u_t \mid \mathbf{z}), \tag{5}$$

where $T$ is determined by $z$, the input to the GRU.

The next improvement to this 83.50% accuracy was achieved by introducing pseudo-convolutional policy gradients (PCPG) in [112] (No. 4, Table 3). The main pipeline was similar to previous works that used spatial temporal convolutions, followed by a residual NN (the authors used ResNet-18 instead of ResNet-34) and Bi-GRU. The model front-end

used CNN to capture short-term spatio-temporal patterns, while the back-end used an RNN to model long-term global patterns. However, the main novelty is the introduction of a new loss function to train seq2seq models for automatic lip-reading. This resulted in a balanced combination of the conventional cross-entropy loss function with PCPG, and the final loss was calculated as follows [112]:

$$L_{combine} = (1 - \lambda) \times L_{CE} + \lambda \times L_{PCPG}, \tag{6}$$

where $\lambda$ is a scalar weight to balance the two loss functions. The cross-entropy loss $L_{CE}$ is a popular way to learn the model, and at each time step, it is calculated as follows [112]:

$$L_{CE} = -\log p(c_1, c_2, \dots, c_U) = -\sum_{u=1}^{U} \log p(c_u \mid c_1, c_2, \dots, c_{u-1}), \tag{7}$$

where $c_u = 1, 2, \dots, C$ is the predicted class label index at time step $u$, and $C$ is the number of categories to be predicted at each time step. The proposed PCPG-based loss is calculated as [112]:

$$L_u = L_{original} = -R_u \times \log P(y_u \mid \times; \theta),$$

$$L'_u = L_{mapping} = L_{u-|k/2|:u+|k/2|} \times w = \sum_{i=u-|k/2|}^{u+|k/2|} w_i \times L_u,$$

$$L_{PCPG} = \sum_{u=1}^{U} L'_u = \sum_{i=u-|k/2|}^{u+|k/2|} \sum_{u=1}^{U} w_i \times L_i, \tag{8}$$

where $w$, $k$, $\theta$, and $p\theta$ stand for the kernel weights, the kernel size, the parameter distribution of the model, and the parameters. $r_u$ and $R_u$ are the immediate reward at time step $u$ and the future expected reward at time step $u$, respectively. Given the reward $r_u$ at each time step $u$, the future expected reward $R_u$ at the current time step $u$ would be computed as $R_u = \sum_{i=u}^{U} \gamma^{U-i} r_i$, where $\gamma$ denotes the discount factor and $U$ is the maximum length of the character sequence. The final reward for the whole sequence $R$ is $R = \sum_{u=1}^{U} R_u$, which is used to denote the cumulative reward of the whole prediction result from the beginning to the end $U(\mathbf{y}_{1:U})$.

In Ref. [113], the authors proposed a deformation-flow-based two-stream network (DFN) (No. 5, Table 3) that achieved 84.13% accuracy. The general idea of the proposed network was to generate deformation flows that could capture information about the movement of faces and learn with self-control. The authors also provided additional lip-reading tips and used bidirectional knowledge distillation loss to teach two streams simultaneously.

The input data for the DFN were a pair of frames (source and target). The result of the DFN was a deformation field, which is a two-channel map of the same size as the input frames. The authors introduced knowledge distillation loss, which is defined as [113]

$$L_{KD}(q_t, q_s) = -\sum_{i=1}^{N} q_t^{(i)} \log q_s^{(i)}, \tag{9}$$

where $q_t$ and $q_s$ denote the soft probability distributions of the teacher's network and the student's network, respectively, and $N$ denotes the number of classes. Its bidirectional version is calculated as follows [113]:

$$L_{BiKD}(q_g, q_d) = L_{KD}(q_g, q_d) + L_{KD}(q_d, q_g) q_s^{(i)}, \tag{10}$$

and the final objective function of the two-stream network is [113]:

$$L = L_{CE}(z_g, y) + L_{CE}(z_d, y) + \lambda L_{BiKD}(q_g, q_d), \tag{11}$$

where $L_{CE}$ stands for the standard cross-entropy loss for classification tasks, $y$ denotes the one-hot vector indicating the word class label of the video, and $\lambda$ is a hyper-parameter indicating the weight of $L_{BiKD}$.

Yet another improvement to this 84.4% accuracy was achieved in Ref. [114] (No. 6, Table 3) by introducing SpotFast networks, a variant of the SOTA SlowFast networks for action recognition instead of 3D CNNs as a front-end feature extractor. It uses a time window as a point path and all frames as a fast path. As a backend, the authors used Transformers with extended memory to study sequential features for classification.

In Ref. [115], the authors achieved a lip-reading accuracy of 84.41% (No. 7, Table 3) by applying the mutual information maximization technique to the standard 3D CNN spatio-temporal front-end, followed by an 18-layer residual network (ResNet-18) and Bi-GRU as the back end.

The key idea of the method was to use local mutual information maximization (LMIM) and global mutual information maximization (GMIM). The local version helps the model to focus more on related spatial areas at each time step and gives more distinctive features. Optimization for LMIM can be described as binary cross-entropy loss [115], as follows:

$$L_{(LMIM)} = E_{p(F,Y)}[\log(LMIM(f,y))] + E_{p(F)p(Y)}[\log(1 - LMIM(f,y))], \qquad (12)$$

This method maximizes global mutual information about the global representation obtained by the Bi-GRU, followed by an additional LSTM along with a linear layer based on the output data from the front-end. This additional LSTM assigned different weights $\beta$ ($T \times 1$-dimensional) for different frames according to the target word. The objective function can be defined as [115]

$$L_{(GMIM)} = E_{p(O,Y)}[\log(GMIM(o,y))] + E_{p(O)p(Y)}[\log(1 - GMIM(o,y))], \qquad (13)$$

Combining the cross-entropy losses with the optimization functions *LMIM* and *GMIM*, the final objective loss function for the whole model is [115]

$$L_{total} = -\sum_{i=1}^{C} Y_i \log \hat{Y}_i - L_{(LMIM)} - L_{(GMIM)}, \qquad (14)$$

where the first term is the cross-entropy loss and $Y_i$ is the label.

In Ref. [116] (No. 8, Table 3), the authors achieved 84.80% lip-reading accuracy. The authors used a two-stage speech recognition model. In the first stage, the target voice was separated from the background noise with the help of appropriate visual information about lip movements. Thus, the authors were the first to use LRW audio data to improve lip-reading performance. In the second stage, the audio modality was again combined with the visual modality in order to better understand speech with a subnet. Pseudo-3D residual convolution was also introduced as an external interface for the model along with Update 1 in ResNet for temporary convolutional networks (TCN). Finally, the authors presented a recurrent unit with blockable attention by elements (EleAtt-GRU), which is more efficient than a Transformer in long sequences.

In Ref. [117] (No. 9, Table 3), the authors achieved 85.00% lip-reading accuracy by making use of unlabelled unimodal data. Front-ends were trained on large-scale unimodal datasets and then integrated into a larger multimodal structure that learned to recognize parallel AV data and translate it into symbols using a combination of connectionist time classification (CTC) and seq2seq decoding. Both components, inherited from single-module learning with self-control, interacted well with each other, as a result of which the multi-modal structure gave competitive results thanks to fine-tuning.

The CTC loss assumes conditional independence between each output prediction and has a form of [117]

$$p_{\text{CTC}}(\mathbf{y} \mid \mathbf{x}) \approx \prod_{t=1}^{T} p(y_t \mid \mathbf{x}), \qquad (15)$$

where $\mathbf{x} = [\mathbf{x}_1, \ldots, \mathbf{x}_T]$ is the input sequence of the frames of the input of the encoder in the fusion module and $\mathbf{y} = [\mathbf{y}_1, \ldots, \mathbf{y}_L]$ are the targets, where $T$ and $L$ stand for the input and target lengths, respectively.

On the other hand, the autoregressive decoder removes this assumption by carrying out direct evaluation a posteriori based on a chain rule that has the form [117]

$$p_{\text{CE}}(\mathbf{y} \mid \mathbf{x}) = \prod_{l=1}^{L} p(y_l \mid y_{<l}, \mathbf{x}), \tag{16}$$

The overall function is calculated as follows [117]:

$$\mathcal{L} = \lambda \log p_{\text{CTC}}(\mathbf{y} \mid \mathbf{x}) + (1 - \lambda) \log p_{\text{CE}}(\mathbf{y} \mid \mathbf{x}), \tag{17}$$

where $\lambda$ stands for the relative weight between the CTC loss and seq2seq loss in the hybrid CTC/attention mechanisms.

In Ref. [118] (No. 10, Table 3), the authors achieved 85.02% lip-reading accuracy. The authors were the first to look into the effects of different facial regions, including the mouth, the whole face, the upper face, and even the cheeks, on the accuracy of lip-reading. They introduced a cutout method that allowed more discriminative features to be learned by maximizing the quality of visual information.

In Ref. [119] (No. 11, Table 3), the authors achieved 85.30% lip-reading accuracy. The authors addressed the limitations of the conventional bidirectional GRU model and proposed changes that further improved its performance. The Bi-GRU layers were replaced with TCN. The authors also introduced a new training procedure that enables training using one single stage.

In Ref. [120] (No. 12, Table 3), the authors achieved 85.40% lip-reading accuracy. The authors presented a multimodal AV bridge structure that can use both audio and visual information, even with a single-modal input. The memory network created an associative bridge between the source and target memory, which took into account the relationship between the two memories. By studying the relationship through an associative bridge, the proposed bridge structure was able to obtain target modal representations within the memory network even with the initial modal input.

In Ref. [121] (No. 13, Table 3), the authors achieved 85.50% lip-reading accuracy, followed by 88.40% lip-reading accuracy (No. 14, Table 3) using new training strategies. These methods were based on observations that a complex NN, together with several customized training strategies, usually obtains high performance on different CV tasks. This was the first work that started to use cascades of NNs in order to boost lip-reading accuracy. By only introducing some easy-to-obtain refinements to the baseline pipeline, the authors obtained an obvious improvement in performance to 88.40%.

In Ref. [122] (No. 15, Table 3), the authors achieved 88.50% lip-reading accuracy. The authors proposed a series of innovations that improved upon existing NN architectures by introducing a depthwise separable temporal convolutional network (DS-TCN) that significantly reduced the computational cost to a fraction of the original model. The proposed lightweight architecture was able to achieve SOTA performance with a comparably low computational cost. The DS-TCN network also utilized knowledge distillation. The idea was to minimize the combination of cross-entropy loss ($\mathcal{L}_{\text{CE}}$) for hard targets and Kullback–Leibler divergence loss ($\mathcal{L}_{\text{KD}}$) for soft targets [122]:

$$\mathcal{L} = \mathcal{L}_{\text{CE}}(y, \delta(z_s; \theta_s)) + \alpha \mathcal{L}_{\text{KD}}(\delta(z_s; \theta_s), \delta(z_t; \theta_t)), \tag{18}$$

Here, $y$ are labels, the parameters of the student and teacher models are $\theta_s$ and $\theta_t$, respectively, and the predictions of the student and teacher models are $z_s$ and $z_t$, respectively. $\delta$ stands for the softmax function, and $\alpha$ is a hyperparameter for balancing loss conditions.

In Ref. [123] (No. 16, Table 3), the authors achieved 88.50% lip-reading accuracy by implementing multi-head visual-audio memory (MVM), which was trained using AV data

sets and remembered audio representations by modeling the relationships of AV representations. In the second stage, only visual input could extract the stored audio representation from memory by examining the studied relationships. Thus, the lip-reading model could supplement insufficient visual information with extracted sound representations.

In Ref. [13] (No. 17, Table 3), the authors achieved 88.70% lip-reading accuracy using a cascade of NNs with strong stages of data augmentation and pre-processing. In the first step, the authors used the Vosk voice activity detection model (https://github.com/alphacep/vosk-api, accessed on 18 May 2023) to extract the voiced part of speech and remove silence from the video material. In the subsequent processing stage, the MediaPipe FaceMesh algorithm [131] was used to detect the mouth region in each frame. After cropping the region of interest (ROI), several procedures were applied, including (1) greyscale conversion, (2) image normalisation, and (3) histogram alignment. This was followed by the MixUp augmentation method to reduce overfitting [132].

Label smoothing (LS) [133] was applied to the labels of those frames that did not have MixUp [13]. The resulting images were formed into packets and loaded into convolutional layers to extract visual features [13]. Given the input sample belonging to the word class $i$, we denote $p_i$ as a prediction logistician and $y$ as an annotated word label [13]. The initial cross-entropy loss is calculated as follows [13,121]:

$$L = \sum_{i=1}^{N} q_i \log(p_i) \left\{ \begin{array}{l} q_i = 0, y \neq i \\ q_i = 1, y = i \end{array} \right., \tag{19}$$

When applying LS, $q_i$ is changed ($\epsilon$ is a small constant) to [13,121]:

$$q_i = \left\{ \begin{array}{l} \epsilon/N, y \neq i \\ 1 - \frac{N-1}{N}\epsilon, y \neq i \end{array} \right., \tag{20}$$

In Ref. [124] (No. 18, Table 3), the authors achieved 89.52% lip-reading accuracy by introducing a resource-efficient E2E architecture, Efficient-Nets. The authors also removed a max-pool layer from the existing SOTA 3D front-end and improved back-end robustness by adding a Transformer encoder.

In Ref. [125] (No. 19, Table 3), the authors achieved 94.10% lip-reading accuracy, which is the current maximum achieved in the scientific literature on the LRW dataset to date. The authors proposed time masking (TM) augmentation followed by MixUp and DC-TCN. The authors also showed that the inclusion of self-distillation and word boundary indicators also provided some benefits, although to a lesser extent.

In Ref. [11] (No. 20, Table 3), the authors achieved 98.56% AVSR accuracy. This is the only AV LRW result available to date. The authors trained a 3DCNN+BiLSTM architecture for lip-reading and used an MLF-based pre-trained model for acoustic speech recognition. They tried three types of modality fusion, prediction-level, feature-level, and model-level, in order to achieve SOTA AVSR results.

### 3.2. LRS2-BBC, LRS3-TED: Audio-Visual Speech Recognition and Fusion Strategies

In a previous section, we described the recent advances in visual speech decoding based on the current lip-reading benchmarking dataset, LRW. However, according to the conducted analysis, the most popular datasets among researchers for AVSR benchmarking are the LRS2-BBC [83] and LRS3-TED datasets [84].

### 3.2.1. LRS2 Dataset Methodology and Results

The AV speech word error rate (*WER*) results for the LRS2-BBC dataset are presented in Table 4. With the introduction of the audio modality (which is generally recognized much better than the visual modality) the speech recognition accuracy is significantly improved.

Thus, researchers tend to use the $WER$ metric on AV tasks since it is commonly used in conventional ASRs. $WER$ is defined as [134]:

$$WER = (S + D + I)/N,\qquad(21)$$

where $S$ is the number of substitutions, $D$ is the number of deletions, $I$ is the number of inserts to be received from the reference to the hypothesis, and $N$ is the number of words in the reference.

**Table 4.** AVSR results on LRS2-BBC dataset in ($WER$, %).

| No. | Method | Year | Video | Audio | Audio-Visual |
|---|---|---|---|---|---|
| 1 | TM-Seq2seq [83] | 2018 | 48.3 | 9.7 | 8.5 |
| 2 | TM-CTC [83] | 2018 | 54.7 | 10.1 | 8.2 |
| 3 | CTC/Attention [135] | 2018 | 50.0 | 8.2 | 7.0 |
| 4 | LF-MMI TDNN [136] | 2020 | 48.86 | 6.7 | 5.9 |
| 5 | Hybrid Conformer [137] | 2021 | 39.1 | 3.9 | 3.7 |
| 6 | MoCo + wav2vec [117] | 2022 | 43.2 | 2.7 | 2.6 |
| 7 | CTC/Attention + Extra training data [138] | 2023 | 14.3 | 1.5 | 1.5 |

The results presented in the table quite accurately reflect the state of the field of research to date. In general, it can be seen that the recognition accuracy of audio-based systems is much higher than the accuracy of visual-based recognition. However, for 6 out of 7 systems, the simultaneous use of two modalities made it possible to achieve the maximum $WER$ result. Below, we will try to explore the main stages in the development of AVSR based on the LRS2 dataset, which made it possible to come close to 100% speech recognition accuracy on such a complex and challenging dataset.

The first AVSR SOTA results were reported in [83] (No. 1, Table 4) and set the bar high with 8.5% $WER$ for AVSR. The visual-only results were only around 50% $WER$ at the time of release of the LRS2 dataset. However, the fusion of both audio and visual modalities resulted in a 1.2% $WER$ gain in comparison with a pure audio-based recognition system.

The authors came up with a sequence-to-sequence transformer (TM-Sec2sec) model architecture. This is a coder-decoder attention structure for seq2seq-style learning. Two input streams, one for video and one for audio, were used in the proposed model. For the acoustic representation, the authors used 321-dimensional spectral features calculated with a window of 40 ms and an overlap of 10 ms at a sampling frequency of 16 kHz. Since the video was sampled at a frequency of 25 frames per second (40 ms per frame), each input video frame corresponded to 4 vectors of acoustic features. The authors cropped an $112 \times 112$ ROI covering the mouth region and applied a 3D ResNet for the extraction of spatio-temporal visual features.

To integrate audio and visual information, the authors employed an internal attention encoder. This encoder consists of stacked multi-headed self-attention layers, where the input tensor serves as the query, key, and value for attention. Separate encoders were utilized for the audio and visual modalities. Fixed positional embeddings in the form of sinusoidal functions were employed to convey the input order information within the model.

For decoding, the authors utilized TM-Sec2sec. At each decoder level, the resulting video and audio contexts were combined along the channel dimension and propagated to the direct communication unit. The decoder generated symbol probabilities that directly corresponded to the ground-truth labels and was trained using cross-entropy loss.

The same authors proposed the TM-CTC architecture in [83] (No. 2, Table 4) and immediately surpassed their previous result for AV speech recognition. It is worth mentioning that novel CTC-based methods are inferior to TM-Sec2sec in terms of uni-modal speech recognition. However, their ability to fuse two modalities led to an overall improvement of 0.3% $WER$.

The basic concept of the TMC TC model is the fusion of video and audio encodings distributed over a series of self-monitoring/preemption blocks. The outputs of the NN are

posterior CTC probabilities for each input frame, and the entire stack was trained using CTC loss. Apart from these aspects, the architecture is very similar to the TM-Sec2sec described above.

In Ref. [135] (No. 3, Table 4), the first significant improvement to the SOTA was achieved. The authors used a 3DCNN followed by an 18-layer ResNet for feature extraction from the visual modality and 80 logarithmic Mel functions, together with pitch, delta pitch, and sound probability (total 83 functions) for the sound modality. Audio functions were extracted using a Hamming window of 25 ms in 10 ms increments.

The main novelty is the use of Hybrid CTC/Attention and two types of modality fusion. A stack of bidirectional LSTMs is used in the encoder to convert input streams $x = (x_1, \ldots, x_T)$ into frame-by-frame representations of hidden functions. These functions are then passed to the decoder, including a RNN language model, attention mechanisms, and CTC, to output a sequence of labels $y = (y_1, \ldots, y_L)$. To align these input frames with output characters, the authors implemented a location-based attention mechanism that considered both content and positional details. During training, the objective function was calculated as a linear combination of CTC and attention, which is calculated as follows [135]:

$$\mathcal{L} = \alpha log p_{ctc}(\mathbf{y} \mid \mathbf{x}) + (1 - \alpha) log p_{att}(\mathbf{y} \mid \mathbf{x}), \tag{22}$$

where $\alpha$ controls the relative weight in the CTC and attention mechanisms. When decoding, a joint CTC/attention approach was used. This approach eliminated the disadvantages of the attention-only approach, which has a non-monotonic alignment and exhibits problems with detecting the end of a sentence. The recognition hypothesis $\hat{y}$ is computed as follows [135]:

$$\hat{\mathbf{y}} = \arg\max_{\mathbf{y} \in \mathbf{U}} \{\lambda log p_{ctc}(\mathbf{y} \mid \mathbf{x}) + (1 - \lambda) log p_{att}(\mathbf{y} \mid \mathbf{x})\} \tag{23}$$

where $\lambda$ is the CTC weight and $U$ s a set of labels plus an additional label at the end of the sentence. The RNN-LM used for decoding was enabled by shallow fusion [139], which can be described as follows [135]:

$$log p^{hyb}(\mathbf{y} \mid \mathbf{x}) = \lambda log p_{ctc}(\mathbf{y} \mid \mathbf{x}) + (1 - \lambda) log p_{att}(\mathbf{y} \mid \mathbf{x}) + \beta log p_{RNN-LM}(\mathbf{y}),$$
$$\hat{\mathbf{y}}^{\star} = \arg\max_{\mathbf{y} \in \mathbf{U}} \left\{log p^{hyb}(\mathbf{y} \mid \mathbf{x})\right\}, \tag{24}$$

where $\beta$ stands for the relative weight of the RNN-LM model.

Another useful decision was the use of both the early and late fusion of AV modalities. In early fusion, audio and visual functions were combined inside the encoder, followed by the use of a Bi-LSTM, which created a common hidden representation fed into the decoder. In a late merge, audio and video were processed independently by unimodal encoder–decoder architectures, and then combined as follows [135]:

$$log p^{hyb}_{late\_fusion} = \gamma log p^{hyb}_{audio} + (1 - \gamma) log p^{hyb}_{visual}, \tag{25}$$

where $\gamma$ ranges from 0.0 to 1.0 and is a hyperparameter for controlling the relative weight between audio and visual probabilities.

In Ref. [136] (No. 4, Table 4), the authors proposed modality fusion gates to robustly fuse the audio and visual features and lattice-free discriminative criterion (LF-MMI) time-delay NN (TDNN) architecture that established new SOTA results for the LRS2 dataset with 6.7% *WER*.

The authors explored three different fusion modality methodologies: feature concatenation, visual-modality-driven gated fusion, and AV-modality-driven fusion. For feature

concatenation fusion, the acoustic features were combined with visual features extracted using LipNet [126]; then the combined features were transferred to RecogNet [140]:

$$p(\mathbf{y_t} \mid \mathbf{x_t}) = \text{RecogNet}([\mathbf{x_t}, \text{LipNet}(\mathbf{v_t})]), \tag{26}$$

where $y_t$ is the alignment at the frame level of the corresponding acoustic frame $x_t$, and $v_t$ is the ROI of the target speaker's mouth.

For synchronized fusion controlled by the visual modality, acoustic characteristics and visual characteristics were transmitted to the visual and audio networks, respectively [136]. Then, the outputs of the audio network were gated by the outputs of the $m_t$ visual network with element-by-element multiplication [136]:

$$\begin{aligned} \mathbf{m_t} &= \mathbf{VisualNet(v_t)}, \\ \mathbf{h_t} &= \text{AudioNet}(\mathbf{x_t}), \otimes \sigma(\mathbf{m_t}), \\ p(\mathbf{y_t} \mid \mathbf{x_t}) &= \text{RecogNet}(\mathbf{h_t}), \end{aligned} \tag{27}$$

where $\otimes$ denotes the Hadamard product and $\sigma$ is a sigmoid function. The audio and visual networks have a similar architecture, with 6 TDNN-F layers each.

For AV-modality-driven controlled fusion, a fusion network was added to the gate structure controlled by a visual modality. The outputs of the audio and video networks were combined and fed into the fusion network before being used at the gating stage [136]:

$$\mathbf{m_t} = \text{FusionNet}([\text{VisualNet}(\mathbf{v_t}), \text{AudioNet}(\mathbf{x_t})]), \tag{28}$$

where the *FusionNet* was represented by a TDNN network containing 3 levels. The *FusionNet* can use both distorted audio and visual information and provide more efficient information for the gating step [136].

In Ref. [137] (No. 5, Table 4), the authors demonstrated that E2E training, the use of a conformer instead of an RNN, and the use of a language model based on a converter significantly improved the basic results for AV recognition, achieving a *WER* of 3.9%.

The AV model encoder consists of three components: an external interface, an internal interface, and fusion modules. For the visual flow, the authors used a modified ResNet-18, in which the first convolutional layer was replaced by a three-dimensional convolutional layer [137]. Visual features at the end of the residual block were compressed in the spatial dimension by the global average pooling layer [137]. For the audio modality, the authors used ResNet-18 based on one-dimensional convolutional layers, where the filter size on the first convolutional layer was 80 (5 ms) [137]. At the end of the external module, the frame rate of the acoustic features was reduced to 25 frames per second to match the frame rate of the visual stream [78].

The authors suggested using a conformer encoder as a server part for time modeling [141]. It consisted of an implementation module followed by a set of conformer blocks. In each block of the conformer, the direct communication module, the multi-head self-monitoring module (MHSA), the convolutional module, and the direct communication module were arranged in order.

The fusion of acoustic and visual features was carried out by a multilayer perceptron and projected to n-dimensional space. The multilayer perceptron consisted of a linear layer with an output size of 4 × n, followed by a batch normalization layer, ReLU, and the last linear layer with an output size of n.

The authors used a hybrid CTC/attention architecture [142] to ensure monotonous alignment and, at the same time, remove the assumption of conditional independence. CTC losses assume conditional independence between each output prediction and have the form of $p_{\text{CTC}}(\mathbf{y} \mid \mathbf{x}) \approx \prod_{t=1}^{T} p(y_t \mid \mathbf{x})$. The attention-based model removes this assumption

by directly estimating the a posteriori probability based on a chain rule that has the form $p_{\text{CE}}(\mathbf{y} \mid \mathbf{x}) = \prod_{l=1}^{L} p(y_l \mid y_{<l}, \mathbf{x})$. The fused function is computed as follows [142]:

$$\mathcal{L} = \alpha \log p_{\text{CTC}}(\mathbf{y} \mid \mathbf{x}) + (1 - \alpha) \log p_{\text{CE}}(\mathbf{y} \mid \mathbf{x}), \tag{29}$$

where $\alpha$ stands for the relative weight of the CTC and attention mechanisms.

In Ref. [117] (No. 6, Table 4), the authors succesfully established new SOTA AVSR results for LRS2 (2.7% *WER*) by significantly increasing the amount of training data. The idea was that audio and visual interfaces are trained on large single-modal datasets. The authors then integrated the components of both interfaces into a larger AV infrastructure that learned to translate parallel AV data into characters using a combination of CTC and seq2seq decoding. The proposed AV model consisted of four components: an external and internal interface for both modalities, a fusion module, and decoders.

The most recent achievement on the LRS2 dataset was 1.5% *WER*, as reported in Ref. [138] (No. 7, Table 4). The authors investigated the use of automatically generated transcriptions of unlabeled datasets to increase the size of the training set [138]. They trained the ASR, VSR and AVSR models on the augmented training sets of several existing datasets, such as LRS2, LRS3, and VocCeleb2. The proposed model achieved new SOTA performance on AVSR on LRS2 and LRS3 and proved that increasing the size of the training set leads to reduced *WER* despite using noisy transcriptions.

The proposed Auto-AVSR leveraged pre-trained models to produce automatically generated transcriptions for unlabelled AV datasets [138]. Audio waveforms from the unlabelled AV datasets were consumed by a pretrained ASR model to produce automatic transcriptions. Then, the authors adopted the architecture proposed in [137] and achieved a new SOTA result.

3.2.2. LRS3 Dataset Methodology and Results

The AV speech *WER* results for the LRS3-TED dataset are presented in Table 5. It is the largest AV speech dataset available for researchers to date. In addition, in comparison to the LRS2 dataset, which is subject to some restrictions and available only for research purposes, the LRS3 dataset is completely free for use by researchers and industry. Some of the methods presented in Table 5 (No. 1, 4, 7) have already been described in the LRS2 section because the authors evaluated their methods on two datasets at once. Therefore, in this section, we will not go deep into their description but will provide the achieved recognition results. Instead, we will focus on the methods we have not previously described.

**Table 5.** AVSR results on LRS3-TED dataset in (*WER*, %).

| No. | Method | Year | Video | Audio | Audio-Visual |
|-----|--------|------|-------|-------|--------------|
| 1 | TM-Seq2seq [83] | 2018 | 58.9 | 8.3 | 7.2 |
| 2 | RNN-T [143] | 2019 | 33.6 | 4.8 | 4.5 |
| 3 | EG-Seq2seq [116] | 2020 | 57.8 | 7.2 | 6.8 |
| 4 | Hybrid-Conformer [137] | 2021 | 43.3 | 2.3 | 2.3 |
| 5 | RAVEn Large [144] | 2022 | 23.4 | 1.9 | 1.4 |
| 6 | AV-HuBERT Large [145] | 2022 | 26.9 | 1.6 | 1.4 |
| 7 | CTC/Attention + Extra training data [138] | 2023 | 19.1 | 1.0 | 0.9 |

In Ref. [83] (No. 1, Table 5), the authors achieved the first baseline AVSR results on the LRS3-TED dataset with 7.2% *WER*. However, the initial VSR model was not so impressive, with a *WER* of only 58.9%, due to the difficult visual conditions of the dataset. Despite the rather poor VSR results, the combined AV model outperformed the ASR model by 1.1% *WER*.

In Ref. [143] (No. 2, Table 5), the authors achieved new SOTA performance of 4.5% *WER* by creating a large AV dataset of segmented utterances extracted from YouTube public videos and using it for training data augmentation. Along with this, the authors proposed the use of an RNN-T model for AVSR.

In Ref. [116] (No. 3, Table 5), the authors proposed a modification to the previous baseline and introduced the EG-Seq2seq model, which achieved 6.8% *WER* on the LS3-TED

dataset. The method was built upon a double visual awareness multi-modality speech recognition (AE-MSR) network. In the first stage, it was trained to select spectrograms based on the acoustics of other speakers or background noise with additional information derived from the visual modality. The subsequent multi-modality speech recognition network then once again passed through the visual modality and fused it with a filtered acoustic spectrogram.

Magnitude spectrograms from the waveform signal at a sample rate of 16 kHz were used as audio features. To align with the video frame rate, the window length of the fast Fourier transform was set to 40 ms, and the hop length was set to 10 ms, corresponding to a 75% overlap. The visual features were obtained by a ResNet-like 3D CNN. In the MSR sub-network, the authors employed an encoder-to-decoder (Enc2Dec) mechanism based on a variant of the RNN model named element-wise-attention GRU (EleAtt-GRU) [146]. This model assigns attention to each element or dimension of the input, allowing for the modeling of different importance levels.

In Ref. [137] (No. 4, Table 5), the authors introduced conformers for AVSR and achieved a *WER* of 2.3%. The methodology was previously described in the LRS2 section and was also benchmarked on the LRS3-TED dataset.

In Ref. [144] (No. 5, Table 5), the authors presented RAVEn, a self-supervised multi-modal approach to jointly learn visual and acoustic speech representations. The proposed architecture aimed to achieve self-supervised multimodal learning by combining the visual and acoustic speech representations. It involved pre-training by encoding masked input data and predicting contextualised targets using evolving pulse encoders. The design of the method was asymmetric, with the audio stream predicting both visual and acoustic events, and the visual stream predicting acoustic events only. By training the visual and audio encoders together in the pre-training phase, RAVEn [144] achieved significant results in both low- and high-resource settings. In particular, RAVEn [144] outperformed all existing self-trained methods for VSR on the LRS3 [84] dataset.

The authors also proposed a loss structure that reflects asymmetry between the audio and visual modalities (the acoustic modality contained more information relevant to speech than the visual one). Denoting the set of mask token indices for audio as $M_a$, the audio-to-audio prediction loss and cross-modal loss, respectively, can be expressed as [144]

$$
\begin{aligned}
\mathcal{L}^{a \to a} &= - \sum_{n \in M_a} \mathrm{sim}\left( f_p^{a \to a}(f_e^a(\bar{x}^a))_n, \mathrm{sg}(g^a(x^a)_n) \right), \\
\mathcal{L}^{m_1 \to m_2} &= - \sum_n \mathrm{sim}\left( f_p^{m_1 \to m_2}(f_e^{m_1}(\bar{x}^{m_1}))_n, \mathrm{sg}(g^{m_2}(x^{m_2})_n) \right),
\end{aligned}
\tag{30}
$$

where $m_1, m_2 \in \{v, a\}, m_1 \neq m_2$, and *sg* denotes the "stop-gradient" operation, which indicates that no gradient is passed back to the teacher networks. At each iteration, the objectives for the video and audio students are [144]

$$
\mathcal{L}_v = \mathcal{L}^{v \to a}, \quad \mathcal{L}_a = \mathcal{L}^{a \to v} + \mathcal{L}^{a \to a}
\tag{31}
$$

For fine-tuning, the authors kept the pre-trained student encoders and discarded the rest.

In Ref. [145] (No. 6, Table 5), the authors presented a self-supervised AVSR framework and methodology built upon Audio-Visual HuBERT (AV-HuBERT), a SOTA AV speech representation learning model. It achieved 1.4% *WER* on the LRS3-TED dataset.

AV-HuBERT is an approach that is used for pre-training the joint representations of audio and video information streams in an unsupervised manner. It takes frame-wise audio and video data as input and generates contextualized AV representations for each frame. AV-HuBERT consists of two main stages: feature clustering and masked prediction. Audio-based MFCC features are usually used for the generation of clusters during the first iteration. Depending on the final architecture that we want to train, the approach can

involve a linear layer for an encoder-only model or a randomly initialized decoder for a sequence-to-sequence model [145].

The most recent evolution of the masked autoencoders was presented in [147]. The Contrastive Audio-Visual Masked Auto-Encoder (CAV-MAE) method combines contrastive learning and masked data modeling, two major self-supervised learning frameworks, to learn a joint and coordinated AV representation [147].

In Ref. [138] (No. 7, Table 5), the authors produced automatically generated transcriptions for unlabelled AV datasets to augment the training data and appled the architecture proposed in [137] to achieve a new SOTA result on the LRS3-TED dataset of 0.9% *WER*.

## 4. Discussion

This section presents the analysis of the surveyed VSR methodologies. Then, it highlights recent improvements in AVSR methodologies.

Through DL algorithms, both VSR and AVSR have made significant advances in the last 10 years. Figure 1 demonstrates the evolution of methodologies for visual speech decoding. We divide recent VSR advances into three main parts: core, pre-processing, and additional techniques.

As we can see from Figure 1, the evolution of the core of the recognition system happened as follows. The initially proposed model did not undergo conceptual changes for a long period of time. It consisted of two main parts, a front-end and a back-end (despite the E2E architecture consisting of one NN, part of it is a front-end module for extracting informative features, and the back-end part is responsible for recognition/classification). There has been significant progress in improving each of the parts while maintaining a similar structure. The initial improvement of the front-end has replaced the 2D convolution with a spatio-temporal 3D version that is better suited for processing video frame sequences. After that, the combination of 2D and 3D CNNs also demonstrated improvements in recognition accuracy.

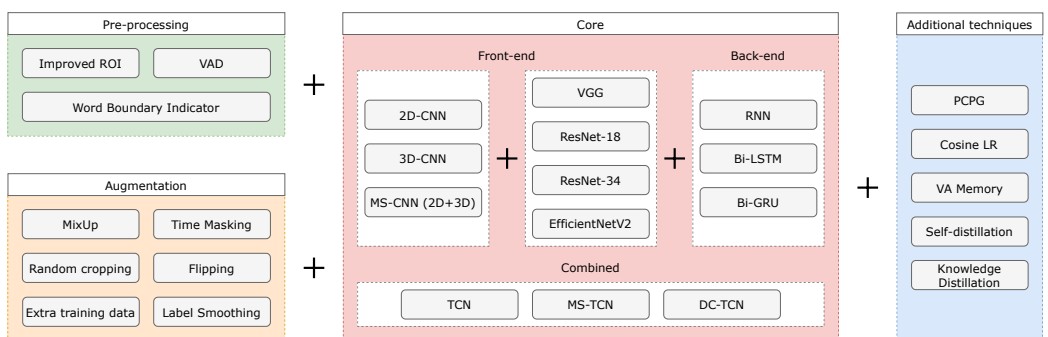

**Figure 1.** Recent evolution of VSR methodologies.

Another important front-end innovation was the use of various NN architectures, which have proven to be effective in many other CV tasks. The VGG, ResNet and Efficient-Net architectures turned out to be especially useful in feature extraction for automated lip-reading tasks. The back-end part has evolved in conjunction with the general CV field. The original RNNs were soon replaced by LSTMs and GRUs and their bidirectional versions.

An important milestone in the development of architectures for VSR systems was the introduction of TCNs to the field. The main disadvantage of a combined front–back-end approach is the need for two separate models. TCN offers a unified approach to cover both levels of information in a hierarchical manner.

The input and output have identical lengths. The field of a TCN is determined by the kernel sizes and the stride. In case of a standard TCN, the activations in each separate layer have the same receptive field [119]. A multi-scale version of TCN was proposed (MS-TCN) to provide the network with visibility in different temporal scales. MS-TCN achieves this

by using multiple branches with different kernel sizes during the feature coding phase, allowing the combination of short-term and long-term information.

Finally, the most recently proposed densely-connected temporal convolutional networks (DC-TCN) are currently the best temporal model for the lip-reading of isolated words. DC-TCN architectures extend the vanilla TCN by adding a dense connection at each TC block and using a squeeze-and-excitation (SE) attention mechanism [125].

Notwithstanding the abovementioned improvements, the rest of the core of modern VSR systems has remained seemingly untouched. However, the most significant increases in accuracy are due to the introduction of some pre-processing stages and various additional techniques that can be adapted to almost any VSR system in order to improve its performance.

Below, we will describe pre-processing techniques that do not directly affect the architecture of the recognition system but can significantly improve the efficiency of speech recognition, and therefore, we consider them promising. All pre-processing techniques can be divided into two main types. The first ones are aimed at more accurate extraction of the ROI from video recordings, and the second ones are augmentation techniques.

The first category includes the use of voice activity detection (VAD) in order to better cut the speech boundaries and remove silence/noises and the use of some improved ROI detection algorithms (e.g., MediaPipe) that are aimed at the more accurate extraction of lip regions from video frames and the use of word boundary indicators. The binary vectors serve as indicators. They have the same length as the input video sequence. Usually, vector inputs that correspond to the frames are set to 1 for the cases where the target word is present, and the rest are set to 0. The word boundary indicator vector is concatenated frame-wise by the encoder and then propagated into the temporal model [125].

The second category includes augmentation techniques, the most promising of which are MixUp, where the new augmented training examples are created by linearly combining two input video sequences and their corresponding targets; time masking, where consecutive frames for each training sequence are masked using a uniform distribution and each masked frame is replaced with the mean frame of the sequence it belongs to; random cropping, where the patch of mouth ROI is randomly cropped during training; flipping, where all the frames in a video are randomly flipped with some probability; LS, which is a regularization technique that well addresses both the overfitting and overconfidence of the trained model (it is usually applied to the labels of those frames that did not have MixUp [13]); and extra training data, which have been proven to be the most effective by adding additional training data from other AV datasets, demonstrating once again that automatic lip-reading along with the CV field is a highly data-dependent task. The best lip-reading result to date was achieved by combining the training data of the four largest publicly available AV speech datasets in Ref. [125].

Some additional techniques that have proven to be advantageous in lip-reading tasks are also depicted in Figure 1 and include several learning functions and strategies, such as PCPG and the cosine learning rate (cosine LR) scheduler, as well as some multimodal knowledge distributions strategies, such as multi-head visual-audio memory, and distillation techniques, such as self-distillation and knowledge distillation.

Self-distillation [148] is built upon the assumption that training a series of models sharing similar architectures by using distillation will benefit automatic lip-reading [122]. It includes a first network, which is a teacher for a student network (both share a similar structure). The student network becomes the teacher network in the next generation. This process continues until the moment that no further improvement is achieved. Knowledge distillation forces models to share inter-class similarity information. It was initially proposed as a transfer learning technique for compression purposes, i.e., when the student capacity is much smaller than the teacher's. However, its applications in AVSR have also provided a great benefit to this field of research.

Thus, following the evolution of speech recognition methods over the past 10 years, we have managed to obtain a set of promising methods in three areas: the recognition

core, pre-processing/augmentation, and additional techniques that improve recognition accuracy. Next, we will try to describe the evolution of AV methods, focusing on the main changes in the recognition core and the way information from different modalities is fused.

Figure 2 demonstrates the methodology evolution for AV speech decoding. We divide recent AVSR advances into four main parts: core, augmentation, and fusion strategies and additional techniques.

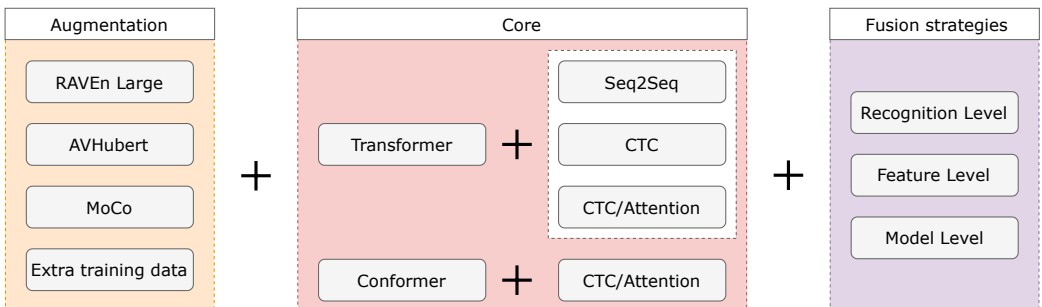

**Figure 2.** Evolution of recent AVSR methodologies.

As we can see from Figure 2, the number of studies on AVSR is much smaller than on automatic lip-reading, and the methodologies have changed less rapidly over the past decade. However, we can follow the recent developments and highlight the major milestones achieved in this field of research. For the most part, E2E DL approaches for AVSR can be divided into two types: the NN as an emission model and sequence-to-sequence models. The first outputs the likelihood of each symbol based on the incoming AV information. The second type reads all of the input sequences first, followed by predicting the final recognition hypothesis.

The first attempt at leveraging recent DL advances for the AVSR task has been the Transformer+Seq2seq model. In this variation, different attention mechanisms are built upon the audio and video embeddings. The resulting video and audio contexts are concatenated in every decoder layer with the channel-wise dimension and followed by passing them to the conventional feedforward block. In both the audio and video modalities, the attention mechanism takes as input the output of the previous decoding layer (or the decoder's input for the first layer). The decoder generates probabilities for each character, which are then matched to the ground-truth labels and trained using cross-entropy loss [83].

Improvements have been made by introducing CTC loss in combination with a Transformer NN. The Transformer-CTC model combines the AV feature representations and passes the result through a number of self-attention/feedforward NN modules. The result is the CTC posterior probabilities for every input video frame. The whole network is trained with CTC loss. CTC-based or Seq2Seq approaches are usually built upon RNNs. However, recently, there has been an upgrade towards completely CNN E2E models.

The next milestone was the use of CTC loss and its fusion with an attention-based trained NN (CTC/attention), with the main goal of simultaneously forcing alignments and removing the conditional independence assumption [135]. In order to map a set of input audio or video streams sequences to corresponding output sequences, a hybrid CTC/attention architecture was proposed. This architecture employs a conventional encoder–decoder structure with an attention mechanism. A stack of Bi-LSTMs is combined in the encoder to convert the input acoustic and visual modality sequences into frame-wise hidden feature representations. These features are then consumed by a joint decoder, including RNN attention and CTC mechanisms, to output a label sequence.

The most recent success to the core of modern E2E AVSRs is the introduction of a convolution-augmented transformer (Conformer). The main idea is that the audio and video information encoders are trained to extract features directly from raw video data (pixels) and acoustic data (waveforms). This stage is followed by them being fed

to conformers, and then fusion takes place with a simple feedforward NN. The use of a conformer instead of an RNN significantly improves the performance of speech recognition.

Another important part of modern AVSR systems is the modality fusion strategy they use. Traditionally, these strategies have been divided into early, late, or hybrid fusion based on the time the fusion happened. However, with recent developments in DL methodologies, researchers started to rely on different classifications, including prediction-level, feature-level, and model-level fusion. Prediction-level fusion is the easiest strategy to implement at its core. From each modality, we obtain a vector of predictions, which is then analyzed by a prediction fusion model. Feature-level fusion demands the use of supplementary models to learn feature relationships both within a single acoustic or video modality and within combined AV modalities. This is the main difference between this fusion strategy and the previous one. In the case of feature-level fusion, both conventional and NN models are commonly used. Finally, in NN-model-level fusion, one single model is trained. The two audio and video models are combined together, and their weights are jointly initialized and fine-tuned. This strategy mimics the work of the human brain, which is able to simultaneously analyze visual and acoustic clues [11]. This strategy has proven to be the most effective in modern AVSR tasks.

Along with this, extra training data are proven to be very effective in almost all modern NN learning related tasks, and AVSR is not an exception. Several data augmentation and training strategies have been recently proposed for AVSR tasks. For example, RAVEn [144] is a multi-modal approach that is trained in a a self-supervised manner to simultaneously learn visual and acoustic speech representations. Pre-training consists of encoding masked inputs and then predicting the contextualized targets generated by slowly evolving momentum encoders. AV-HuBERT [145] is a SOTA AV speech representation learning model. It receives the input of frame-wise audio and video data in order to create contextualized AV representations for each timestep (video frame). The AV-HuBERT pre-training procedure consists of two stages: feature clustering and masked prediction. MoCo is another successful example of applying transfer learning based on a large MoCo v2 visual dataset [117].

## 5. Conclusions

We have reviewed the recent advances in the field of AVSR. They are classified and described in relation to several important research questions in this area. We have also provided detailed information on the latest AV speech datasets. In addition, we presented our view of the evolution of the field of research over the past decade. Finally, we discussed the remaining issues and presented our understanding of future research.

We conclude this article by summarizing our views on where modern AVSR stands and where it is likely heading. Despite all the existing progress we have covered in this article, i.e., the basic methodology of visual and AVSR, AV datasets, pre-processing and augmentation techniques, and modality fusion strategies, there is still much to be done on these topics. DL undoubtedly improves the performance of AVSR, as it does in every other area it has covered. It has just begun to be used for AVSR, but the initial results that have already been obtained are very positive and encouraging.

As mentioned earlier, although there are a number of AV datasets, it is likely that none of them has all the desired characteristics, such as a sufficient amount of data, real-life variability, conventional experiment setups, etc. This limits progress in the field. Recent advances have made better use of data that exist "in the wild", such as on YouTube, BBC video, etc. Since much of these data are unlabeled, deep and multi-view learning can be effective for future data collection. With DL, data representation can be explored without the need to manually develop new feature sets.

Based on the conducted analysis, we can conclude that there is ample room for improvement in the AVSR field, and there is still a lack of innovative approaches, methods, and technologies to advance the SOTA and make existing speech recognition systems more convenient and user-friendly to a global audience.

**Author Contributions:** Conceptualization, D.I. and D.R.; formal analysis, D.R.; methodology, D.I.; investigation, D.I. and D.R.; writing—original draft preparation, D.I. and D.R.; visualization, D.I.; supervision, A.K.; project administration, D.R.; funding acquisition, A.K. All authors have read and agreed to the published version of the manuscript.

**Funding:** This study was supported by the RFBR (Project No. 19-29-09081) and partially supported by a grant (No. MK-42.2022.4), by the Leading Scientific School (NSH-17.2022.1.6), as well as by a state research grant (Topic No. FFZF-2022-0005).

**Institutional Review Board Statement:** Not applicable.

**Informed Consent Statement:** Not applicable.

**Data Availability Statement:** Not applicable.

**Conflicts of Interest:** The authors declare no conflict of interest.

## Abbreviations

The following abbreviations are used in this manuscript:

| | |
|---|---|
| 3D | 3-dimensional |
| AE-MSR | Awareness multi-modality speech recognition |
| ASR | Automatic speech recognition |
| AV | Audio-visual |
| AV-HuBERT | Audio-visual HuBERT |
| AVSR | Audio-visual speech recognition |
| BN | Batch normalization |
| CAV-MAE | Contrastive audio-visual masked auto-encoder |
| CNN | Convolutional neural network |
| CTC | Connectionist temporal classification |
| CV | Computer vision |
| Conformer | Convolution-augmented transformer |
| Cosine LR | Cosine learning rate |
| DFN | Deformation-flow-based two-stream network |
| DL | Deep learning |
| DS-TCN | Depthwise separable temporal convolutional network |
| E2E | End-to-end |
| EleAtt-GRU | Element-wise attention gated recurrent unit |
| Enc2Dec | Encoder-to-decoder |
| GMIM | Global mutual information maximization |
| GRU | Gated recurrent unit |
| LF-MMI | Lattice-free discriminative criterion |
| LMIM | Local mutual information maximization |
| LRW | Lip-reading in the wild dataset |
| LS | Label smoothing |
| LSTM | Long short-term memory |
| MFCC | Mel-frequency cepstral coefficients |
| MHSA | Multi-head self-attention |
| ML | Machine learning |
| MLP | Multi-layer perceptron |
| MVM | Multi-head visual-audio memory |
| NN | Neural network |
| PCPG | Pseudo-convolutional policy gradient |
| RNN | Recurrent neural network |
| ROI | Region-of-interest |
| ReLU | Rectified linear units |
| SOTA | State-of-the-art |
| STCNN | Spatio-temporal convolutional neural network |
| TCN | Temporal convolutional networks |
| TDNN | Time-delay neural network |
| TM | Time masking |

| TM-Sec2sec | Sequence-to-sequence transformer |
| VAD | Voice activity detection |
| VSR | Visual speech recognition |
| WER | Word error rate |

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
