# Peer review of "A Review of Recent Advances on Deep Learning Methods for Audio-Visual Speech Recognition"

_mathematics, doi:10.3390/math11122665_

Round 1
Reviewer 1 Report
This paper reviewed the recented results for AVSR over the last decde. AV datasets and methodology were discussed in this paper. A discussion on the current state of AVSR and future research were provided.
This paper reviewed the recented results for AVSR over the last decde. AV datasets and methodology were discussed in this paper. A discussion on the current state of AVSR and future research were provided.
Author Response
We sincerely thank the reviewer for your valuable time and efforts in reviewing our manuscript. Those comments are all valuable and very helpful for revising and improving our article, as well as the important guiding significance to our researches. We have studied comments carefully and have made correction which we hope meet with approval.
The description of different fonts used in this document are as follows:
- Reviewers’ original comments are reproduced in red-colored
- Plain fonts are our answers to Reviewers’ comments.
- Text reproduced from the article is shown in blue color.
Comment 1: This paper reviewed the recented results for AVSR over the last decde. AV datasets and methodology were discussed in this paper. A discussion on the current state of AVSR and future research were provided.
Reply: We sincerely appreciate your thoughtful review of our article. Your review is invaluable and we are grateful for your time and expertise. Thank you for your support.

Reviewer 2 Report
There is no description of the symbols used in the formulas. Add a chapter with abbreviations and variables.
I propose to allow this paper after minor changes.
Author Response
We sincerely thank the reviewer for your valuable time and efforts in reviewing our manuscript. Those comments are all valuable and very helpful for revising and improving our article, as well as the important guiding significance to our researches. We have studied comments carefully and have made correction which we hope meet with approval.
The description of different fonts used in this document are as follows:
- Reviewers’ original comments are reproduced in red-colored
- Plain fonts are our answers to Reviewers’ comments.
- Text reproduced from the article is shown in blue color.
Comment 1: There is no description of the symbols used in the formulas. Add a chapter with abbreviations and variables.
Reply: In our article, we faced the challenge of integrating several formulas from different studies in a clear and coherent way. The lack of a consistent combination of variables in these formulas had the potential to confuse readers and introduce ambiguity. To address this issue, we decided to improve the clarity and accessibility of the article by including a reference to the original source in the form of a scientific article before each formula.
By including these references, readers will be able to easily locate and consult the specific articles in which each formula was originally introduced. This additional information will enable the reader to gain a deeper understanding of the context and derivation of the formula, thereby reducing any potential confusion or ambiguity. Our intention is to empower readers by providing them with the necessary resources to explore the underlying research behind the formulas presented.

Reviewer 3 Report
This paper provide a comprehensive view of aduio-visual speech recognition (AVSR). They first reviewed the public datasets in the AVSR area, and then evaluated many deep learning based methods, and finally discussed the remaining issues. During the presentation, they also classified the datasets and methods and summarized them in the table or the figure of methodology, which make the review very clear. It is much better if the datasets can be listed by the order of year or their size in the table. When this is revised, this paper is good to go.

Yes, it's very clear to me.
Author Response
We sincerely thank the reviewer for your valuable time and efforts in reviewing our manuscript. Those comments are all valuable and very helpful for revising and improving our article, as well as the important guiding significance to our researches. We have studied comments carefully and have made correction which we hope meet with approval.
The description of different fonts used in this document are as follows:
- Reviewers’ original comments are reproduced in red-colored
- Plain fonts are our answers to Reviewers’ comments.
- Text reproduced from the article is shown in blue color.
Comment 1: It is much better if the datasets can be listed by the order of year or their size in the table.
Reply: We have successfully sorted the datasets in Table 1 and the methods in Table 5 by year. The remaining tables (2, 3, 4) were also sorted in the initial version of this paper.
Line 123: Table 1.
Line 742: Table 5.

Reviewer 4 Report
This paper presents the review of DL based audio-visual speech recognition.
l Strength:
This paper is well-written and easy to follow.
And the paper provides a solid theoretical analysis about AVSR in the era of DL.
l Weakness
This paper does not deal with AVSR before DL.
The paper lacks recent literature.
l Suggestion
According to weakness 1, It would be better if there was a brief introduction to AVSR before DL and an explanation of the difference in performance.
According to weakness 2, please add recent research trends.
For Example, MAE(Masked Auto-Encoder) has been showing good performance recently.
Such as the following paper that applies MAE to AVSR(from a slightly different point of view): Contrastive Audio-Visual Masked Autoencoder | OpenReview (ICLR 2023)
Author Response
We sincerely thank the reviewer for your valuable time and efforts in reviewing our manuscript. Those comments are all valuable and very helpful for revising and improving our article, as well as the important guiding significance to our researches. We have studied comments carefully and have made correction which we hope meet with approval.
The description of different fonts used in this document are as follows:
- Reviewers’ original comments are reproduced in red-colored
- Plain fonts are our answers to Reviewers’ comments.
- Text reproduced from the article is shown in blue color.
Comment 1: This paper does not deal with AVSR before DL.
Reply: We appreciate the comment of the reviewer. We acknowledge that our article lacks a description of the AVSR methodology prior to deep learning (DL). However, our aim was to focus on the significant advances in the field over the last decade, which happened to be primarily associated with DL techniques.
There are already numerous comprehensive reviews of the “traditional” approaches that preceded DL, and we have tried to avoid duplicating these works in order to keep the length of the article reasonable. Nevertheless, we have included relevant explanations in response to this comment.
Line 70: This paper is focuses on the most recent advances in audio-visual (AV) speech decoding. Significant progress has been witnessed in this field due to the recent boom of DL. Our main intention is to supplement (but not replace) previous studies [5, 72–74]. We propose to look at the problem of AVSR from the DL point of view. We do not consider so-called “traditional approaches” associated with hand-crafted features, Markov models used for recognition [73], etc. And we focus more on modern end-to-end (E2E) AVSR systems and related methodology.
Comment 2: The paper lacks recent literature.
Reply: We appreciate the comment of the reviewer. Almost all sources of literature were written in the last 10 years and more than half in the last 5 years. We regret that this did not seem to be enough recent. However, for our part, we have made every effort to present the current state of the field of research. Due to the specifics of the review, most of the methodology appeared in the 16-20s. therefore, in the article it was given more attention than the methods of 21–23, but they are also described in this review.
Comment 3: It would be better if there was a brief introduction to AVSR before DL and an explanation of the difference in performance.
Reply: We appreciate the comment of the reviewer. We have added a brief introduction of the AVSR preceding deep learning. Due to the significant superiority of the latter on modern databases, a direct comparison could not be shown. However, the absolute superiority of deep learning methods over traditional approaches has been proven in several existing reviews, and we refrained from repeating these experiments due to the limited volume of the article.
Line 29: In order to improve the accuracy and reliability of ASR systems in the late 90s, in addition to analyzing the information from only acoustic modality, researchers began to use video information about speech. In many research works, the developed audio-visual speech recognition systems (AVSR) have demonstrated better recognition results [6–9] than their unimodal counterparts. Fusing streams of audio and visual modalities significantly increases the efficiency of ASR (i.e., the principle of synergy is respected).
Line 54: Despite the significant advances in DL that have been made in recent years in a wide range of fields, including computer vision (CV) (semantic segmentation [18–22], scene understanding [23–27], pose estimation [28–34], action [34–36] or gesture [37–41] classification, face [42–45] or emotion [46–49] recognition, etc.), natural language processing (text analysis [50–52], language translation [53–55], sentiment analysis [56–58], question answering [59], etc.), speech recognition [60–63], and generative design (automated content generation [64], creative image synthesis [65–67], artistic style transfer [68–70], etc.), the development of AVSR methodology is still at an early stage and does not yet meet the performance standards required for practical implementation in real-world applications.
Comment 4: Please add recent research trends. For Example, MAE (Masked Auto-Encoder) has been showing good performance recently. Such as the following paper that applies MAE to AVSR (from a slightly different point of view): Contrastive Audio-Visual Masked Autoencoder | OpenReview (ICLR 2023).
Reply: We appreciate the comment of the reviewer. Indeed, Masked Auto-Encoder is a very promising solution and we considered several research papers that use masks to improve audio-visual speech recognition, such as:
- Haliassos, A.; Ma, P.; Mira, R.; Petridis, S.; Pantic, M. Jointly Learning Visual and Auditory Speech Representations from Raw Data. ArXiv 2022, abs/2212.06246. https://doi.org/10.48550/arXiv.2212.06246.
- Shi, B.; Hsu, W.N.; Mohamed, A. Robust Self-Supervised Audio-Visual Speech Recognition. Interspeech 2022, pp. 2118–2122. https://doi.org/10.21437/interspeech.2022-99.
We would like to thank the reviewer for bringing this article to our attention. We acknowledge that it was published on 11 April 2023, which was towards the end of our writing process for this review. We greatly appreciate the reviewer's efforts to help us improve our methodological framework, and as a result, we have included a reference link to the mentioned article in our revised version.
Line 799: The most recent evolution of the masked autoencoders has been presented in the work [145]. Contrastive Audio-Visual Masked Auto-Encoder (CAV-MAE) method combines contrastive learning and masked data modeling, two major self-supervised learning frameworks, to learn a joint and coordinated audio-visual representation.

Reviewer 5 Report
Abstract: Change "introduction" to "the introduction" in the 5th line. The article should include the definite article "the" before the noun "introduction". We recommend inserting "the" in this sentence.
Introduction: Add a comma after "applications" in the 48th line. A comma should be used in sentences with dependent clauses. We recommend inserting one here. In the 61th line, the word "is fous" should be changed to "focuses" to match the semantic context.
Benchmarking datasets: Change "contain" to "containing" in the 143th line to correct the participle error.
"Speech is the most natural, convenient and understandable way for people to communicate." The original sentence "ASR is the most natural, convenient and understandable way for people to communicate" should be corrected as ASR stands for Automatic Speech Recognition system and is not a general term for human speech communication.
Adding graphs appropriately when discussing different methods in the Methods chapter can achieve better results.
There is not much content related to machine learning in the article, so it is more appropriate to remove machine learning from the keyword section.
Line 110 syntax error ‘play’ should be changed to ‘plays’.
In table2, there is a lot of data in the table. It is recommended to insert dividing lines between rows and rows in the table, so that it may make your table look more clear.
line 591 The first letter of the word ‘belo’ should be capitalized
Line 606 'In order to... Encoder. " do not conform to the grammar, the information may be added after",".
In summary, I was happy to review your manuscript:”A Review of Recent Advances on Deep Learning Methods for Audio-Visual Speech Recognition”. Overall, it is clear that a lot of work has gone into this research.The quality of the article is good. Overall Recommendation is Accept after minor revision(Minor editing of English language required).
Minor editing of English language required
Author Response
We sincerely thank the reviewer for your valuable time and efforts in reviewing our manuscript. Those comments are all valuable and very helpful for revising and improving our article, as well as the important guiding significance to our researches. We have studied comments carefully and have made correction which we hope meet with approval.
The description of different fonts used in this document are as follows:
- Reviewers’ original comments are reproduced in red-colored
- Plain fonts are our answers to Reviewers’ comments.
- Text reproduced from the article is shown in blue color.
Comment 1: Change “introduction” to “the introduction” in the 5th line. The article should include the definite article “the” before the noun “introduction”. We recommend inserting “the” in this sentence.
Reply: Thank you very much! We have taken this comment into account and made the necessary corrections to the article.
Comment 2: Add a comma after “applications” in the 48th line. A comma should be used in sentences with dependent clauses. We recommend inserting one here. In the 61th line, the word “is fous” should be changed to “focuses” to match the semantic context.
Reply: Thank you very much! We have taken this comment on board and made the necessary corrections to the article.
Comment 3: Benchmarking datasets: Change “contain” to “containing” in the 143th line to correct the participle error.
Reply: Thank you very much! We have made the necessary corrections to the article in accordance with the review comment.
Comment 4: “Speech is the most natural, convenient and understandable way for people to communicate.” The original sentence “ASR is the most natural, convenient and understandable way for people to communicate” should be corrected as ASR stands for Automatic Speech Recognition system and is not a general term for human speech communication.
Reply: You are definitely right! Corrections have been made to the article.
Comment 5: Adding graphs appropriately when discussing different methods in the Methods chapter can achieve better results.
Reply: We agree with the reviewer’s comment. Of course, the use of graphs can increase the visibility of the article. However, due to the fact that the volume of the article has already significantly exceeded the original limits, we decided to use tables for this purpose. Even if they lose visibility, we believe that they provide a fairly complete description of the available state-of-the-art methodological information.
Comment 6: There is not much content related to machine learning in the article, so it is more appropriate to remove machine learning from the keyword section.
Reply: We have taken the reviewer’s comment into account and removed the term machine learning from the keywords section.
Comment 7: Line 110 syntax error “play” should be changed to “plays”.
Reply: Thank you very much! We have made the necessary corrections to the article in accordance with the comment.
Comment 8: In table2, there is a lot of data in the table. It is recommended to insert dividing lines between rows and rows in the table, so that it may make your table look more clear.
Reply: We made the necessary corrections to Table 2 according to the comment.
Comment 9: Line 591 The first letter of the word “belo” should be capitalized.
Reply: We have made the necessary corrections to the article in accordance with the comment.
Comment 10: Line 606 “In order to... Encoder.” do not conform to the grammar, the information may be added after”,”.
Reply: We appreciate the reviewer's efforts in proofreading the article. We do not hesitate to take into account any spelling or typographical errors found. We have made the necessary corrections to the article in accordance with the comment.
